# Peer review of "Improvement of Microstructure and Sliding Wear Property of Cold-Sprayed FeAl Intermetallic Compound Coating by Annealing Treatment"

_coatings, doi:10.3390/coatings13071260_

Round 1

Reviewer 1 Report

This manuscript offers an interesting investigation on the microstructure and microhardness of cold-sprayed FeAl intermetallic compound coatings. The authors characterized the samples by means of SEM, X-ray diffraction diffractometer, TEM and Vickers microhardness. The study was carried out for as-cold-sprayed and after annealing samples.

However, in my opinion the paper is too short, and it lacks a mechanical characterization which can really define the effects of the heat treatment. Therefore, the authors should add an abrasion test to their study.

Moreover, the authors should include some practical examples of applications for FeAl intermetallic compound coatings in the introduction section.

The paper can be accepted after major revisions.

Reviewer 3 Report

The article deals with the improvement of the microstructure and microhardness of the cold sprayed FeAl intermetallic compound coating by annealing treatment. This article is a continuation of the authors' research on this topic.

The Introduction sufficiently describes the existing state of knowledge and justifies undertaking the research topic. The authors also refer to their own publications.

Materials, research methods and analysis of results have been included, but need to be supplemented.

Detailed review

1. Based on what studies were the annealing treatment times selected?

2. Line 24: Please remove the bolding of "Cold".

3. The substrate material should be better characterized. In my opinion, the information that "sandblasted stainless steel plates were used as a substrate" (line 94) is insufficient, because it is too general a term.

4. How was the substrate prepared?

5. Line 84: In articles, authors usually include information about the technique used (SEM, TEM) when describing microstructure images. It would be good to include such information in the caption to Fig. 1.

6. Line 112: The authors cite the EDX analysis but do not post its results.

7. Line 116: The authors in Fig. 2b have enlarged part of Fig. 2a. Please indicate in Fig 2a the area from which the photo was taken.

8. Line 120: The explanation of the abbreviation "SAD" used is missing. It would be good to explain it as the authors have done with other abbreviations.

9. Line 134: In Fig. 4, it would be useful to add the test results performed before the annealing treatment.

10. Line 211: Please add a space between the content of the article and the Acknowledgment.

Round 2

Reviewer 1 Report

The authors have made the required updates. In my opinion the paper can be accepted in the present form.